# Electrically reversible cracks in an intermetallic film controlled by an electric field

Z.Q. Liu[1], J.H. Liu[1], M.D. Biegalski[2], J.-M. Hu[3], S.L. Shang[3], Y. Ji[3], J.M. Wang[1], S.L. Hsu[4], A.T. Wong[5], M.J. Cordill[6], B. Gludovatz[7], C. Marker[3], H. Yan[1], Z.X. Feng[1], L. You[8], M.W. Lin[2], T.Z. Ward [5], Z.K. Liu[3], C.B. Jiang[1], L.Q. Chen[3], R.O. Ritchie [4,9], H.M. Christen[2] & R. Ramesh[4,9,10]

Cracks in solid-state materials are typically irreversible. Here we report electrically reversible opening and closing of nanoscale cracks in an intermetallic thin film grown on a ferroelectric substrate driven by a small electric field (~0.83 kV/cm). Accordingly, a nonvolatile colossal electroresistance on–off ratio of more than $10^8$ is measured across the cracks in the intermetallic film at room temperature. Cracks are easily formed with low-frequency voltage cycling and remain stable when the device is operated at high frequency, which offers intriguing potential for next-generation high-frequency memory applications. Moreover, endurance testing demonstrates that the opening and closing of such cracks can reach over $10^7$ cycles under 10-μs pulses, without catastrophic failure of the film.

[1] School of Materials Science and Engineering, Beihang University, Beijing 100191, China. [2] Center for Nanophase Materials Sciences, Oak Ridge National Laboratory, Oak Ridge, TN 37831, USA. [3] Department of Materials Science and Engineering, The Pennsylvania State University, University Park, PA 16802, USA. [4] Department of Materials Science and Engineering, University of California, Berkeley, CA 94720, USA. [5] Materials Science and Technology Division, Oak Ridge National Laboratory, Oak Ridge, TN 37831, USA. [6] Erich Schmid Institute of Materials Science, Austrian Academy of Sciences, and Department of Material Physics, Montanuniversität Leoben, Jahnstr. 12, 8700 Leoben, Austria. [7] School of Mechanical and Manufacturing Engineering, UNSW Sydney, Sydney, NSW 2052, Australia. [8] School of Optical and Electronic Information, Huazhong University of Science and Technology, Wuhan 430074, China. [9] Materials Sciences Division, Lawrence Berkeley National Laboratory, Berkeley, CA 94720, USA. [10] Department of Physics, University of California, Berkeley, CA 94720, USA. Correspondence and requests for materials should be addressed to Z.Q.L. (email: zhiqi@buaa.edu.cn)

Recent efforts on the integration of magnetic intermetallic alloy thin films with functional ferroelectric oxides have created exciting opportunities for manipulating magnetism and resistivity of intermetallics via small electric fields, which offers great promise for low-energy-consuming memory device applications[1–5]. Such heterostructures, such as FeRh/BaTiO$_3$ and FeRh/0.72PbMg$_{1/3}$Nb$_{2/3}$O$_3$–0.28PbTiO$_3$ (PMN-PT), rely on the strong interfacial-strain-mediated magnetoelectric coupling between the piezoelectric effect in the ferroelectric layer and the phase instability around the phase transition of the FeRh layer to generate giant magnetization and resistivity modulation[1,2,4].

In addition, the piezoelectric effect of ferroelectric oxides has been extensively employed in mechanical energy harvesting, mechanical sensors, and motors. Nevertheless, the strength degradation of ferroelectric ceramics under cyclic electric fields often generates cracks due to the microscopic internal stress produced at domain boundaries by electrostriction and domain-switching deformations[6–10], which are detrimental to functionality in most applications. The cracks typically further grow upon cycling of electric fields, thus promoting premature fatigue failures in ferroelectric ceramics. The growth dynamics of cracks in ferroelectric ceramics is indeed closely related to the polarity of the electric field relative to the prepoling field. For example, an opposite electric field relative to the prepoling field enhances the crack propagation, while an electric field of the same polarity inhibits the crack propagation[11]. Also, the occurrence of mechanical opening and closing of cracks under an alternating electric field has been found as the major mechanism for electric-field-induced crack growth in ferroelectric ceramics[12].

Reversible, reproducible, and durable mechanical sensors based on cracks have been demonstrated although cracks were considered as defects to be avoided[13]. More importantly, the crack formation in solid-state materials could be well controlled by bending[13], notches, and confined surface stress[14–16]. In this work, we demonstrate a new type of heterostructures consisting of intermetallic alloys and ferroelectric oxides for room-temperature memory applications; interestingly, we can utilize precisely the otherwise problematic existence of cracks to create a functional device. Instead of utilizing electronic phase transitions of intermetallic alloys and the piezoelectric effect of ferroelectric oxides, here we propose to employ the mechanical properties of intermetallic alloys and the opening/closing of cracks induced by cyclic electric fields in ferroelectric oxides. By integrating moderately ductile intermetallic alloy thin films onto ferroelectric oxide single-crystal substrates, mechanical opening and closing of cracks in ferroelectric oxides driven by external electric fields can be transferred into the intermetallic films via interfacial strain mediation, and then, the ferroelectric oxides effectively act to mechanically open and close fissures in the intermetallic films in a similar fashion to opening/closing an electrical breaker, but here, at the nanoscale, consequently resulting in a colossal on–off ratio of the resistance switching across cracks in the intermetallic thin films.

## Results

**Materials and sample growth**. Intermetallic alloys are a unique class of materials consisting of ordered alloy compounds formed between two or more metallic elements, where different elements occupy specific sites in the crystal lattice. Their mechanical properties are intermediate between metals, which are generally soft and ductile, and ceramics, which are generally hard and brittle. Strikingly, the properties of intermetallics can be strongly influenced by small changes in the system, for example, small variations in the microstructure can result in large changes in strength and ductility[17].

To utilize the opening and closing of cracks in ferroelectric ceramics as electrically detectable switches, intermetallic thin films are ideal candidates to be integrated on top because of their intermediate mechanical properties and resistivity. That is because with a metal thin film grown on a ferroelectric substrate, when cracks form and open in the ferroelectric oxide with application of a cyclic electric field, corresponding cracks in the metal thin film may well not form immediately, or at all, due to the good ductility of the metal layer. Conversely, if a ceramic film is grown on top of a ferroelectric oxide crystal, when the cracks in ferroelectric substrate close, the cracks in the ceramic film may not be fully closed and remain partially open, leading to a high resistance across the crack, making this system highly insulating and prone to premature failure as ceramic materials tend to be extremely brittle. Therefore, intermetallic alloys are precisely the right materials to be considered for such a structure.

MnPt, an antiferromagnetic (AFM) intermetallic alloy, forms a CuAu-I ($L1_0$)-type tetragonal crystal structure with $a = 2.827$ Å and $c = 3.669$ Å. The tetragonal phase is stable with Pt composition between 33 and 60 at%[18], where the lattice constants and the Néel temperature strongly depend on the Pt composition. Our theoretical calculations on the elastic properties of the AFM Mn$_{50}$Pt$_{50}$ reveal that its bulk modulus $B = 177.3$ GPa (Supplementary Note 1) is close to the measured elastic modulus of MnPt films (Supplementary Note 2), shear modulus $G = 96.5$ GPa, and Young's modulus $E = 245.1$ GPa. This yields a bulk/shear modular ratio $B/G \approx 1.84$, indicating that MnPt is likely to be brittle due to the AFM Mn atoms according to Pugh's criterion[19] (Supplementary Note 3). Thus, defect-free MnPt is expected to be very close to the empirical boundary between ductile and brittle. Indeed, our previous electrical measurements demonstrate that MnPt thin films are metallic with room-temperature resistivity $\rho \sim 10^{-4}$ Ω cm[5]. In light of this, we conclude that MnPt represents a possibly ideal candidate material to develop a crack-based memory device in an intermetallic/ferroelectric heterostructure, where the ferroelectric layer effectively acts to mechanically open and close fissures in the MnPt layer to act as the nanoscale analog of a conventional switch.

To realize such a switch, we deposited MnPt thin films onto (001)-oriented PMN-PT single-crystal substrates ($5 \times 5 \times 0.3$ mm$^3$), which were coated with Au on the backside and prepoled by applying a positive bias to the backside Au before deposition. Figure 1 shows a cross-section TEM image of a MnPt/PMN-PT heterostructure. The MnPt layer is polycrystalline with multiple

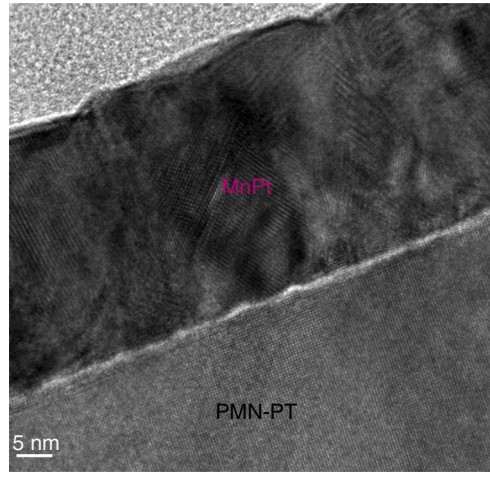

**Fig. 1** Structural characterization. Cross-section transmission electron microscopy of a 35-nm-thick MnPt/PMN-PT heterostructure. The scale bar on the left bottom corresponds to 5 nm

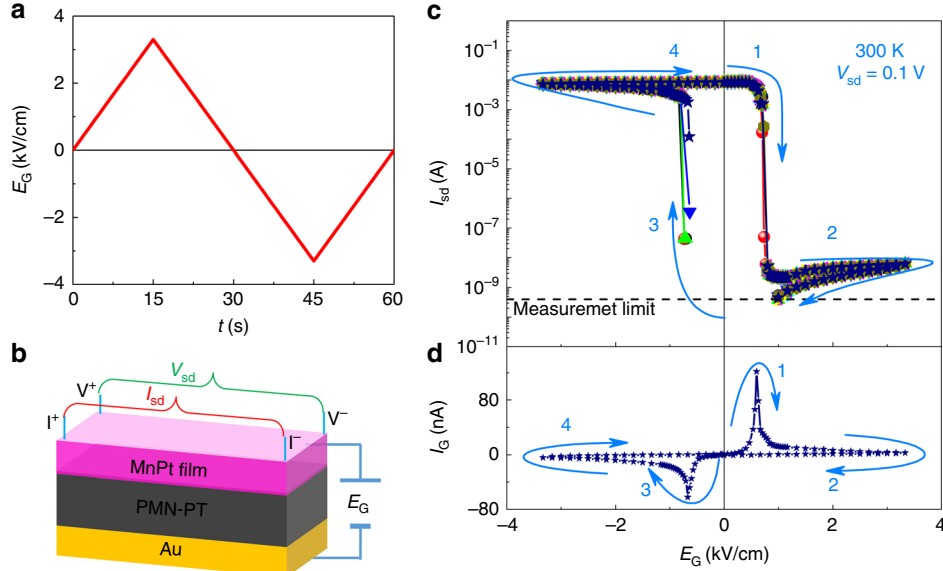

**Fig. 2** Colossal electrical switching at room temperature. **a** Electrical waveform for creating cracks in PMN-PT. **b** Schematic of the resistance measurement geometry. **c** Repeated and reproducible switching of the channel current $I_{sd}$ as a function of the gate electric field $E_G$, measured by $V_{sd} = 0.1$ V at 300 K. The data for nine continuous cyclic field loops are plotted (data connecting segments 2 and 3 are not shown because they are below the measurement limit). **d** Corresponding switching current in the PMN-PT substrate of the last scan. The arrows and numbers in **c** and **d** are the guidance of the field-sweep sequence

orientations. Moreover, the interface is reasonably sharp, with an interface layer that is only ~1-nm thick. Scanning transmission electron microscopy–energy-dispersive spectroscopy (STEM–EDS) analysis revealed that the chemical composition of the MnPt film was $Mn_{44.2}Pt_{55.8}$ ($\pm 2$ at%). The Mn deficiency may be because the lighter Mn atoms are more strongly scattered to higher angles by the background Ar atoms.

**Colossal electroresistance**. To generate cracks in the PMN-PT substrate which would then propagate through the MnPt film as well, a triangular cyclic electric field, with an amplitude of 3.3 kV/cm and a period of 60 s (Fig. 2a), was applied between the top MnPt film and the back Au electrode. After 100 cycles, electric-field-driven reversible opening and closing of cracks with sub-micrometer width and few micrometers length were visible under a microscope. The opening of the cracks occurs when the applied field is opposite to the prepoling field direction, while closure sets in when the external field is reversed. It is important to remind ourselves that ferroelectric crystals do not switch uniformly but rather through the motion of domain walls, which are partially pinned at surface defects. The local strains occurring near a surface are therefore highly nonuniform, and not symmetric with respect to the reversal of the applied field. In our case, this leads to a switching behavior such that a positive bias (segments 1 and 2) results in crack opening, whereas a negative bias (segments 3 and 4) leads to the closing of the cracks, despite the fact that the average strain state of the crystal in both directions is such that one would expect a compressive state—locally, however, domain motion clearly leads to a tensile behavior without which the cracks would not open. Because the metal layer is deposited onto a surface with preexisting domains and pinning defects, these two states are highly reproducible and not symmetric with a field, contrary to the macroscopic strain in the crystal.

Subsequent electrical measurements were performed to examine the effect of the crack opening and closing on the resistance of the top MnPt film. To probe the macroscopic response of an ensemble of cracks, the Van der Pauw geometry was utilized in the initial measurements, which is schematically

represented in Fig. 2b. At $E_G = 0$ kV/cm, a fixed voltage of $V_{sd} = 0.1$ V induces a current of $I_{sd} \sim 10^{-2}$ A, corresponding to a room-temperature resistance of ~10 $\Omega$ (Fig. 2c). Increasing $E_G$ to +0.83 kV/cm drives a sudden drop in $I_{sd}$ over several orders of magnitude to ~$10^{-9}$ A. Accordingly, the switching current in the PMN-PT exhibits a sharp peak (Fig. 2d), characteristic of the ferroelectric domain switching. $I_{sd}$ continues to decrease with increasing $E_G$, and becomes too low to read as we approach $E_G$ to +1 kV/cm. Subsequent reduction of $E_G$ to 0 kV/cm does not change the open state of the measurement circuit, illustrating a nonvolatile high-resistance state.

$I_{sd}$ increases and can be experimentally resolved again as $E_G$ decreases to −0.83 kV/cm. The corresponding peak in the leakage current of PMN-PT shows the opposite domain switching relative to that at +0.83 kV/cm. Above this gate charge, $I_{sd}$ rapidly increases to ~$10^{-2}$ A and saturates. The saturation current remains as the gate bias is removed at $E_G = 0$ kV/cm. This provides a nonvolatile low-resistance state. Finally, we note that the writing and reading signals are applied independently of each other, and reading of the device cannot trigger a back-switching behavior.

**Mechanism for the reversible cracks**. To directly visualize the status of cracks for nonvolatile high- and low-resistance states, atomic force microscopy was utilized to image a MnPt thin-film surface as $E_G$ was cycled from +3.3 kV/cm to 0 kV/cm and from −3.3 kV/cm to 0 kV/cm. As seen in Fig. 3a, a crack is open with a gap width of ~250 nm after sweeping $E_G$ from +3.3 kV/cm to 0, corresponding to the high-resistance open state in the resistance measurements. Conversely, the crack closes after the hetero-structure experiences a negative $E_G$ of −3.3 kV/cm, yielding an electrically intact low-resistance state (Fig. 3b). The depth of the crack is much larger than the thin-film thickness of 35 nm, which is consistent with our interpretation that the cracks are trans-ferred from the ferroelectric oxide into the thin film via interfacial strain. Moreover, the clear correspondence between the resistance switching and the leakage current peaks in Fig. 2 demonstrates that the opening and closing of the cracks in PMN-PT are

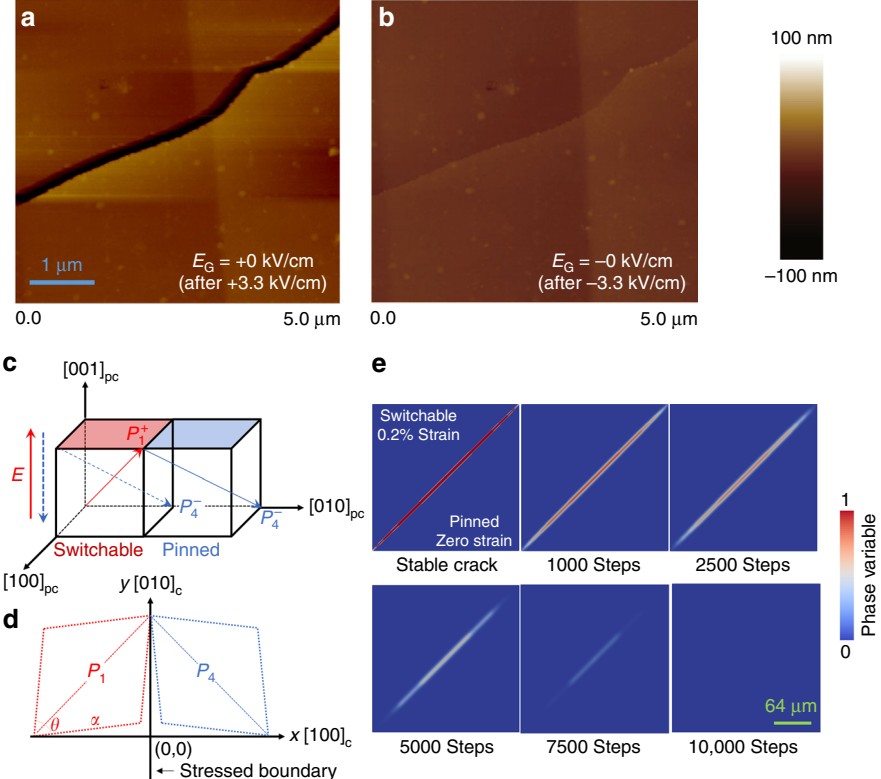

**Fig. 3** Reversible crack driven by an electric field. Atomic force microscopy images (5 × 5 μm²) of a single crack in the MnPt film after scanning the gate electric field $E_G$ **a** from +3.3 kV/cm to 0 kV/cm and **b** from −3.3 kV/cm to 0 kV/cm. **c**, **d** Schematics showing that electric-field (E)-induced reversible and nonvolatile crack formation and closure on the (001) PMN-PT surface can result from a reversible and nonvolatile E-induced 109° polarization switching. A two-domain configuration consisting of one switchable polarization domain and one pinned domain is considered. Such reversible 109° polarization switching can repeatedly induce an in-plane shear strain of about 0.2% in the switchable domain, which will repeatedly stress the domain boundary. The 0.2% strain is calculated based on the lattice parameters (a,θ) of the rhombohedral (001) PMN-PT. **e** Phase-field simulations of the crack evolution when the strain is ON (the first image in the first row) and OFF (the remaining images). Color bar shows the magnitude of the phase variable, which equals 1 in the crack region and 0 in the PMN-PT, and changes continuously across their interface

induced by internal stresses generated at domain boundaries due to domain-switching deformations.

Figure 3c–d shows one possible mechanism for the electric-field-induced crack formation and closure on the surface of PMN-PT. A simplified two-domain configuration is considered, where one polarization domain (left) can be switched by an electric field while the other is pinned (Fig. 3c). In this case, an upward electric-field (E)-induced 109° polarization switching from $P_4^-$ to $P_1^+$ at the surface of (001) PMN-PT crystal (demonstrated by experiments[20]) can cause a local shear strain of about 0.2% in the left domain. This 0.2% shear strain is calculated based on the lattice parameters of the rhombohedral PMN-PT at 300 K[21] (a = 4.017 Å, θ = 89.89°, see Fig. 3d). Such a large strain will cause severe increase in the elastic energy such that a crack forms along the wall to relax the stress, at the expense of producing additional surface energy. Once the $P_1^+$ domain is switched back to the $P_4^-$ domain by applying a downward electric field, the strain returns to zero. The crack will then close to reduce surface energy. Since the E-induced 109° polarization switching has been experimentally demonstrated to be both reversible and nonvolatile[20], the associated crack opening and closing should also be reversible and nonvolatile. Overall, the electric-field-induced crack formation and closure are mainly governed by the competing electromechanical-strain-related elastic energy and the surface energy. The higher the surface energy, the harder the crack formation (i.e., the higher the crack formation energy).

A simplified phase-field model was developed (see "Methods" section) to demonstrate this mechanism. As shown in Fig. 3e, the

crack can remain stable when a 0.2% strain is applied to the top half of the simulation zone (corresponding to the $P_1^+$ domain), while the strain in the other half (corresponding to the $P_4^-$ domain) is set as zero. Once the strain in the entire zone is set as zero (corresponding to a uniform $P_4^-$ domain), the crack will gradually vanish to reduce the surface energy (see the corresponding dynamic evolution process in Fig. 3e).

**High-frequency properties and endurance test.** Given that cracks in a ferroelectric are known to have the potential to result in macroscopic fracture, we further investigate the crack growth dynamics of MnPt/PMN-PT heterostructures under cyclic electric fields. The samples were first excited by 100 cycles of triangular waveform to create cracks. The samples were then mounted into a setup that is similar to what was used in ref. [12], but is capable of measuring smaller samples to monitor the cracking progress operando. As plotted in Fig. 4a, the crack growth as a function of the cycling frequency demonstrates that excitation frequencies above 300 Hz can significantly suppress crack propagation and can withstand a much longer cycling process. The origin of the exponential dependence of the crack lifetime on frequency is not very clear and is still under investigation.

To understand the reason for which cracks grow more slowly at high frequency, we performed measurements using short pulses. It was found that the reversible opening and closing of the cracks can be driven by gate electric pulses as short as 10 μs spaced by 30 ms. We therefore performed the endurance test for a single crack (~300-nm width, ~100-μm length) via two-probe

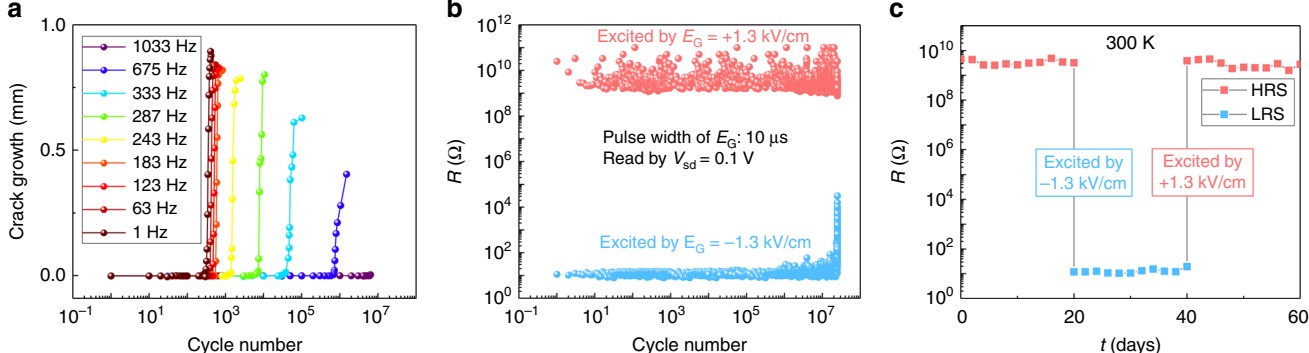

**Fig. 4** Crack growth dynamics and endurance testing. **a** Crack growth length vs. electric cycling number under different cyclic field frequencies. The amplitude of the cyclic electric field with triangular waveform is 1.3 kV/cm. **b** Two-probe resistance measured by 0.1 V vs. gate electric field pulse number across a single crack. The pulses are of 10-μs width and the amplitude is 1.3 kV/cm. **c** Retention of different resistance states of a single crack up to 60 days

resistance measurements. Upon applying $\pm 1.3$ kV/cm and 10-μs electric pulses to the PMN-PT substrate, the resistance across the crack was measured by a small voltage of 0.1 V. The reversible opening and closing of the crack can be cycled more than $10^7$ times before the crack grows rapidly and the closing state degrades (Fig. 4b), which is good for memory applications[22]. This shows that the different behavior at different frequencies is largely related to the amount of time the device is subject to the field around the switching moments, which provides us with flexibility in designing devices for use in different circumstances.

**Memory performance**. Based on our experience on small device fabrication of such heterostructures, the crack state is stable for at least 22 months. Representative data for the retention of the high-resistance and low-resistance states (HRS and LRS) collected in 2 months are shown in Fig. 4c. The excellent nonvolatility is consistent with crack formation as a consequence of the local strain and deformation at domain boundaries induced by non-uniform domain switching. In contrast, the macroscopic strain in PMN-PT has a symmetric butterfly loop and is volatile at zero field[23]. These results demonstrate a prototype memory device based on an intermetallic/ferroelectric heterostructure. It is expected that both the switching speed[24] and the cycling number could be largely increased by patterning the cracks into much smaller scales or further refining the device layout.

**Discussion**

Compared with the intrinsic resistivity modulation of intermetallic alloys by piezoelectric strain in intermetallic/ferroelectric heterostructures, such as FeRh/PMN-PT[2] and FeRh-BaTiO₃ (Ref. 4), where the $\rho$-$E_G$ curves are almost symmetric with "butterfly" shape, the resistance switching in crack-based intermetallic/ferroelectric heterostructures exhibits "square" loops (Fig. 2c). This is a favorable attribute for nonvolatile information storage. Both types of resistance switching predominantly rely on the domain switching of ferroelectrics, and therefore, the switching speed should be intrinsically comparable. However, the former can only be operated around the phase transition temperatures of the intermetallic alloys, and the resistance modulation is limited to the intrinsic resistivity difference between different electronic phases, which, for example, is maximally ~50% for FeRh[25] between the ferromagnetic and AFM phases. In contrast, crack-based switching can be operated over a large temperature range below ferroelectric $T_c$ and is capable of a colossal resistance on−off ratio.

The microstructure and composition of an intermetallic alloy are important for its mechanical properties, such as ductility. Compared with the defect-free $Mn_{50}Pt_{50}$ in our initial theoretical

calculations, the practical films are polycrystalline and have a chemical ratio of $Mn_{44.2}Pt_{55.8}$, which could thus possess a different ductility than theoretically predicted. Although more detailed studies on the mechanical properties of such thin films are needed, the fact that the MnPt/PMN-PT heterostructures exhibit remarkably reversible on and off states suggests that the polycrystalline feature and the current composition of the MnPt films may be important for achieving the excellent closure state with "healed" electrical conductivity.

Such intermetallic/ferroelectric heterostructures are indeed also microelectromechanical systems (MEMS). The crack formation is expected to occur even in 1-μm-thick high-quality PMN-PT films[26], which would lead to an operation voltage of ~0.1 V. In contrast, most of conventional MEMS need an operation voltage of 3−8 V for pulling in counter electrodes[27]. Compared with conventional MEMS that consist of multiple layers and need complicated fabrication processes, the one-layer structure of our devices is much simpler and the fabrication process is much easier. In addition, most of conventional MEMS are volatile at zero voltage while our devices are nonvolatile.

The elastic energy for switching the polarization in ferroelectric oxides can be estimated as $E_{Switching} = 1/2P_sVS$[28], where $P_s$ is the saturation ferroelectric polarization (~25 μC/cm² for PMN-PT), $V$ is the switching voltage, and $S$ is the cell area. Such an energy consumption is predominant for opening and closing the cracks when such a device is of large size. However, as the device is scaled down, the bonding energy of the freshly created surfaces of a crack could be dominant, which can be estimated as the surface energy times the interface area of the crack. For example, if a crack device can be scaled down to a footprint area of $100 \times 100$ nm² with PMN-PT thickness of 1 μm for integrated device applications, the energy consumption for switching the polarization would be $E_{Switching} = 1/2P_sVS = 12.5$ μC/cm² $\times$ 0.83 kV/cm $\times$ 1 μm $\times$ 100 nm $\times$ 100 nm $\approx$ 0.1 fJ. In contrast, considering a typical surface energy value of ~1 J/m², a film thickness of 35 nm, such as that in our case, and a crack length of 100 nm, the surface energy of the two surfaces of the crack is $2 \times 1$ J/m² $\times 35 \times 10^{-9}$ m $\times 100 \times 10^{-9}$ m = 7 fJ, which is much larger than the polarization-switching energy.

In summary, instead of trying to avoid the cracks in ferro-electrics that have been regarded as being highly detrimental in terms of promoting premature fatigue failures, we have utilized cyclic-field-induced cracks in ferroelectric oxides by integrating moderately ductile intermetallic alloy films on top and realized electrically controllable switches with a colossal on−off electro-resistance ratio. Together with the high-frequency properties and the excellent endurance performance, this work illustrates a prototype crack-based memory device. Such an approach could

be feasible for a wide range of intermetallic/ferroelectric hetero-structures, thus opening a new avenue to information storage.

## Methods

**Sample growth.** MnPt thin films were sputtered from a $Mn_{50}Pt_{50}$ target onto (001)-oriented PMN-PT single-crystal substrates ($5 \times 5 \times 0.3$ mm$^3$) in a radio frequency sputtering system with a base pressure of $3 \times 10^{-7}$ Torr. All the PMN-PT substrates were coated with Au on the backside and prepoled by applying a positive bias to the backside Au before deposition. To avoid interfacial oxidation of MnPt and chemical reaction between MnPt and PMN-PT, the deposition temperature was kept at 375 °C. The sputtering power and Ar gas pressure were 40 W and 3 mTorr, respectively. To avoid possible breaking of the PMN-PT substrates during variations in temperature, the ramp rate was kept at 5 °C/min for both heating and cooling. The deposition rate was 0.19 Å/s, as determined by transmission electron microscopy (TEM) measurements. With a deposition time of 30 min, the resulting thin-film thickness of the MnPt was ~35 nm.

**Electrical and structural and mechanical measurements.** Electrical contacts onto MnPt films were made by Al wires via wire bonding. The electrical measurements were performed by Keithley 2400 source meters at room temperature. For the STEM characterization, cross-sectional wedged samples were prepared by mechanical polishing on Allied High Tech Multiprep and ion milled by a Gatan Precision Ion Milling System. By utilizing an electron probe to scan thin films to achieve high resolution of local regions, Z-contrast scanning transmission electron microscope images were acquired by high-angle annular detector on FEI TitanX microscope at 300 kV. For mechanical crack growth measurements, the samples were immersed in silicon oil inside a transparent plexiglass container and subjected to a cyclic electric field through the bolts attached to the electrodes. The crack growth was monitored and measured by a microscopy system and a digit charge-coupled device system. Pulse measurements for cycling endurance testing were performed by an Agilent pulse generator.

**Calculation methods.** The single-crystal elastic constants of MnPt were predicted by the strain–stress method based on first-principles calculations[29], and the elastic properties for polycrystalline MnPt were estimated using the Hill approach[30] (see Supplementary Methods for details).

**Phase-field simulations.** We introduce a continuous phase variable $\eta$ to describe the crack region ($\eta = 1$), the bulk PMN-PT ($\eta = 0$), and their interface (more precisely, the surface of the PMN-PT, where $0 < \eta < 1$). In the framework of diffuse-interface theory, the total free energy $F_{tot}$ of such a two-phase system can be written as:

$$F_{tot}(\eta, \varepsilon) = \int_V \left[ f_{chem}(\eta) + \frac{\kappa_{ij}}{2} \nabla_i \eta \nabla_j \eta + f_{elast}(\eta, \varepsilon) \right] dV, \quad (1)$$

where $\kappa_{ij}$ is the gradient energy coefficient (J/m) that is proportional to both the surface energy $\gamma$ and the surface depth (or interface thickness) of the PMN-PT. As there are no available data on the surface energy of the complex perovskite 0.72PMN–0.28PT, we used the surface energy of (001) BaTiO$_3$ ($\approx$1.076 J/m$^2$ in ref. [31]), a prototypical ferroelectric oxide with perovskite structure, as an approximation. To describe the crack region and the PMN-PT, the chemical free energy density $f_{chem}$ takes a function that has two global energy minima at $\eta = 0$ and 1,

$$f_{chem}(\eta) = w\eta^2(1 - \eta)^2, \quad (2)$$

where $w$ is the potential barrier separating the two energy minima, and is proportional to the ratio of surface energy to the surface depth[32]. The elastic energy density $f_{elast}$ is calculated as:

$$f_{elastic}(\eta, \varepsilon) = \frac{1}{2} \sigma_{ij} \varepsilon_{ij} = \frac{1}{2} c_{ijkl}(\eta) \varepsilon_{kl} \varepsilon_{ij} = \frac{1}{2} c_{ijkl}(\eta) \left( \bar{\varepsilon}_{kl} + \delta\varepsilon_{kl} - \varepsilon^0_{kl} \right) \left( \bar{\varepsilon}_{ij} + \delta\varepsilon_{ij} - \varepsilon^0_{ij} \right), \quad (3)$$

where the spatially variant elastic stiffness tensor $c_{ijkl}(\eta)$ is given by

$$c_{ijkl}(\eta) = c^{PMN-PT}_{ijkl}(1 - h(\eta)) + c^{crack}_{ijkl} h(\eta). \quad (4)$$

Here, the use of the interpolating function $h(\eta) = \eta^3(6\eta^2 - 15\eta + 10)$ makes $c_{ijkl} = c^{PMN-PT}_{ijkl}$ when $\eta = 0$; $c_{ijkl} = c^{crack}_{ijkl}$ when $\eta = 1$, $c_{ijkl}$ is a combination of $c^{PMN-PT}_{ijkl}$ and $c^{crack}_{ijkl}$ at the solid–crack interface. The elastic stiffness tensor of PMN-PT $c^{PMN-PT}_{ijkl}$ is taken from ref. [33], while the $c^{crack}_{ijkl}$ is set $10^{11}$ times smaller than the $c^{PMN-PT}_{ijkl}$ (not exactly zero to avoid numerical issues).

The $\bar{\varepsilon}_{kl}$, $\delta\varepsilon_{kl}$, and the $\varepsilon^0_{kl}$ in Eq. (3) represent the homogeneous strain, heterogeneous strain, and the stress-free strain, respectively. The homogeneous strain $\bar{\varepsilon}_{kl}$ describes the average deformation of the system. It is therefore spatially invariant and is zero in the present system. The stress-free strain $\varepsilon^0_{kl}$ describes the influence of the local polarization on the local strain in a ferroelectric material, and

is given by

$$\varepsilon^0_{kl} = Q_{klmn} P_m P_n, \quad (5)$$

where $Q_{klmn}$ is the electrostrictive stiffness tensor; $P_m$ and $P_n$ represent local polarization vectors. In the two-domain configuration (Fig. 3c), $\varepsilon^0_{12}$ is set as 0.2% in the $P_1^+$ domain and 0 in the $P_1^-$ domain (see "Discussion" section in main text). The heterogeneous strain $\delta\varepsilon_{kl}$ is given by

$$\delta\varepsilon_{kl} = \frac{1}{2} \left( \frac{\partial u_k}{\partial x_l} + \frac{\partial u_l}{\partial x_k} \right), \quad (6)$$

where the spatially variant $u$ is the local mechanical displacement in the system, and is obtained by solving the mechanical equilibrium equation $\partial\sigma_{ij}/\partial x_j = 0$, expanded as

$$c_{ijkl} \frac{\partial^2 u_k}{\partial x_l \partial x_j} = c_{ijkl} \frac{\partial \varepsilon^0_{kl}}{\partial x_j}. \quad (7)$$

Equation (7) is numerically solved using a Fourier-spectral iterative perturbation method[34].

The evolution of the crack is modeled using relaxational kinetics[35,36]

$$\frac{\partial}{\partial t} = -LH(f_{elast} - f_c) \frac{\delta F_{tot}}{\delta}, \quad (8)$$

where $L$ is the mobility of the crack–solid interface, which was set as a constant for simplicity[35,36]. $H(f_{elast} - f_c)$ is the Heaviside step function that equals 0 when $f_{elast} < f_c$ and when $f_{elast} \geq f_c$. Here, the $f_c$ is a parameter for modeling the nucleation and growth of the crack, which can be related to the surface energy through the Griffith relation[36]. Since the surface energy is proportional to the crack formation energy, $f_c$ can be related to the crack formation energy. Indeed, in our present model, the value of $f_c$ ($\approx$10 MJ/m$^3$) is found to be dependent on the surface energy we chose. The variational derivative $\delta F_{tot}/\delta$ represents the thermodynamic driving force for the evolution of the phase variable $\eta$, expanded as

$$\frac{\delta F_{tot}}{\delta} = \frac{\partial(f_{chem} + f_{elast})}{\partial} - \kappa_{ij}\nabla_i\nabla_j. \quad (9)$$

Equation (8) is then solved using a numerically efficient semi-implicit Fourier-spectral method[36] in a 3D discretional grid system of $256\Delta x \times 256\Delta y \times 2\Delta z$, with $\Delta x = \Delta y = \Delta z = 1$ μm. The obtained spatial distribution of $\eta$ at different time steps (Fig. 3e) describes the evolution of the crack morphology.

**Data availability.** The data and simulation codes that support the findings of this study are available from the corresponding author on request.

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

## Acknowledgements

Z.Q.L. acknowledges financial support from the National Natural Science Foundation of China (NSFC Grant No. 51771009) and the start-up grant from Beihang University. This work was partially supported by the Laboratory Directed Research and Development (LDRD) Programs of ORNL managed by UT-Battelle, LLC (sample growth) and the DOE Office of Science, Basic Energy Sciences, and Materials Sciences (partial electrical transport). B.G. and R.O.R. were supported by the U.S. Department of Energy, Office of Science, Basic Energy Sciences, Materials Sciences, and Engineering Division. J.-M.H., Y.Ji, and L.-Q.C. acknowledge financial support from the US National Science Foundation under the DMREF program with grant number DMR1629270. R.R. acknowledges financial support from the NSF (Nanosystems Engineering Research Center for Translational Applications of Nanoscale Multiferroic Systems, Cooperative Agreement Award EEC-1160504). We acknowledge Y. Lee for collecting the AFM data. We dedicate this work to the late Dr Michael D. Biegalski, who was not only a well-respected research colleague, but also a wonderful father and husband, a driven athlete, and a dear friend to all of us.

## Author contributions

Z.Q.L. performed sample growth and electrical measurements with assistance from M.D.B., A.T.W., M.W.L., T.Z.W., J.H.L., J.M.W., H.Y., Z.X.F., L.Y. and J.C.B. S.L.H. performed TEM measurements. M.C., B.G. and R.O.R. contributed to mechanical testing of MnPt films. S.L.S, C.M. and Z.K.L. performed theoretical calculations on elastic properties of MnPt. J.-M.H., Y.J. and L.-Q.C. contributed to understanding the mechanism and the phase-field simulations of strain-mediated electric-field-driven crack formation and closure. This project was coordinated by Z.Q.L., H.M.C. and R.R. All authors contributed to the discussion of results. Z.Q.L. wrote the manuscript with help from all other authors.

## Additional information

**Competing interests:** The authors declare no competing financial interests.

