## [Peer Review File · Nature Communications]

Reviewers' comments:

Reviewer #1 (Remarks to the Author):

1) This paper reports about a novel device type for current switching on the micro scale, based on opening and closing of micro cracks in a (well-chosen) metal layer activated by a piezoelectric thin film (PMN-PT). The switch works with bipolar gate voltages of interestingly low level (about 1/- 1 V), and exhibits over 4 orders of magnitude on/off current ratio, as required by microelectronic logic applications. The authors also showed a lifetime of $10^{(+7)}$ cycles. This is quite a good start. The device has the potential to replace a field effect transistor (FET's), however, needing a bipolar voltage input (positive voltage to be in open state, and a negative voltage to be in closed state, or the other way round). FETs have a defined state at zero gate voltage, and just need one gate polarity to change the status.

2) The current cannot behave the same way as the ferroelectric polarization. A crack opens as soon as the electric field applied to the PMN-PT exceeds a certain threshold value, which is in the general case not coinciding with the ferroelectric switching (as shown in fig. 2c), but it can. As soon as $E_g > E_c$, the stress (strain) flips to a positive value, which potentially can lead to a cracking in the adjacent conductor film. The loop show in fig. 2c does not correspond to a physically logic picture. The stress at branch 2 is as much tensile as the branch 4, and both should be in the cracked, i.e. low current state. This loop is possibly obtained for a situation where a first tensile stress cycle (no 4 in fig. 2c) was not sufficient to crack the line, but the second (no 1 \diamond 2) was. This was accidentally like this. The next cycle would look differently.

3) As to memory applications, the suitability of the crack device is more delicate. It is not shown that the cracking is bistable. Only the ferroelectric loop is it (see above). It is not shown what is going on at zero Volt with a crack. The memory effect requires that the state remains unchanged when the gate field is removed. It is clear that when the state is a closed=uncracked state nothing is changing. However, if we deal with an open=cracked state, how can we be sure that there is no migration to close the crack (there can be a negative strain/stress imposed by ferroelectric domains, even without applied field), so that at small reading fields, a closed state appears. Even more difficult is the situation when a cracked line should be reprogrammed for an un-cracked one. How is it possible to heal the crack, and how long does it take? This is not discussed or shown in the manuscript.

4) Fig.4: did the authors make sure that the applied voltage is really applied on the device? An RC problem might reduce the amplitude at higher frequency.

5) Finally one can say that the shown element can be used as a switch, useful for logic applications. However, the memory case is doubtful. Would need more convincing data on the evolution of the cracked state at zero field and reading field, and a concept for closing the crack. The authors should reduce their claims about the memory, and consider correctly the relation between stress and polarization (see figure in pdf file).

Reviewer #2 (Remarks to the Author):

This is very nice piece of well written work with a clear objective and convincing results. I have no scientific questions. The only missing part is the electromechanical modelling of the reversible crack opening and closure. As the objective of the manuscript is more an engineering issue this might be not necessary.

My main concern is whether the manuscript is suitable for Nature Communications. As I am not working in the semiconductor storage area it is difficult for me to judge that and to estimate the impact of this investigation for this community. By trend I would see it more appropriate in a more engineering related journal.

Reviewer #3 (Remarks to the Author):

In their paper, Electrically Reversible Cracks in Intermetallic Film controlled by an Electric Field the authors study the reversible changes in a MnPt film deposited on a PMN-PT substrate. Such a substrate is known to undergo elastic and inelastic deformation in the presence of an electric field. The authors discover that the former effect, elastic deformation, allows for the reversible transfer of the latter, inelastic deformations (in this case cracks), from the substrate material to the film.

This discovery deserves a great deal of credit and I greatly enjoyed reading the paper and it is my pleasure to recommend it for publication. This is truly a novel switch and the provided results are both convincing and sufficiently well described to allow reproduction by other researchers. However, though the fundamental observations of the paper are sound and I think at a minimum this will provoke people to think much more broadly about the types of possible memories and switches can be achieved.

There is at least one serious issue that must be addressed before the paper can be considered acceptable for publication. There are likewise a few minor issues that must be addressed as well as some further investigation which I encourage the authors to do to strengthen in the claims in the paper.

Firstly, the serious issue concerns the so called "critical value" of $B/G = 1.75$. Quite frankly, until I read this paper, I've never heard of it. It's not mentioned in my texts on fracture mechanics. The reference you have provided, reference 15, is a completely inappropriate reference since it is neither the primary source of this "critical value" or an authoritative reference (such as a review or textbook). Reference 15, however, does apparently cite the origin of this "critical value" and it appears to be from a seminal work by Pugh from 1954.

Modern references constantly talk about "Pugh's criterion" for a ductile-brittle transition based on the B/G ratio and a "critical value", but nowhere in Pugh's 1954 paper does he ever formulate this criterion. The paper isn't about a universal ductile-brittle number, instead, it's about comparing the ductility of similar materials by comparing their relative B/G value. The critical value you reference (sometimes stated as 1.74, 1.75, and otherwise 2) is the B/G ratio of iridium, which happens to be the least ductile in the family of "elemental f.c.c. metals of high melting point" as he calls them in his paper.

This is important, because, as is often discussed in texts in fracture mechanics, many metals, particularly BCC metals, will undergo a brittle-ductile transition with temperature. This is because ductility is determined by the number of active slip systems which allow for the motion of dislocations (in many metals, this depends on temperature). As Pugh makes a point of noting, the B/G ratio doesn't work if you try to compare materials with vastly different melting temperatures. Zinc, for example, has a much lower melting temperature than the "elemental f.c.c. metals of high melting point" and so therefore has a much higher homologous temperature. Its B/G ratio is 1.59, below the so-called universal "critical value" and, according to Pugh, undergoes 25% elongation before fracture during tensile testing. That makes zinc decidedly ductile and Pugh explicitly limits the validity of his model to elemental metals at 1/3 their homologous temperature. Some authors appear to include this

caution explicitly when citing Pugh.

The scope of Pugh's model is further limited by the fact that it doesn't even work for all "elemental f.c.c. metals of high melting point" outside of the transition metals. Thorium meets all the limitations of the model with a B/G ratio of 1.74 (in Pugh's paper), but it undergoes up to almost 50% elongation before fracture. This is similarly true for uranium (though it is orthorhombic). Its B/G ratio is about 1 and is also quite ductile at room temperature.

I have tried to determine the origin of this "critical value" formalism as you use it. In the literature, and the earliest reference I can find is: J. R. Morris, et al., *Acta Materiala*, 52, 4849-4857 (2004). This paper was published in 2004, nearly 50 years after Pugh's criterion was formulated! This paper discusses Pugh's work in better detail and do take steps to limit the scope of critical value to "FCC metals with melting temperatures greater than 900C," but later authors (mostly users of DFT) appear to be continuously expanding the scope of Pugh's criterion to cover almost all materials without critical evaluation as to its origin - this has caused me a great deal of confusion.

If you can disprove this by providing an authoritative reference (preferably one published before 2004) or by showing where in Pugh's paper he proves that this is a generalizable critical value for all materials, I would be happy to defer. However, if you cannot do this, I recommend you reformulate this section based on Pugh's original criteria – that is, among similar materials below 1/3 their homologous temperature, the relative B/G ratio can act as a predictor of brittleness. In which case, since the L10 structure is an FCC derivative and Mn/Pt are both transition metals, you could reasonably compare MnPt to the other "high melting point FCC metals" if it also has a high melting point. Do your theoretical calculations also provide an estimate of the melting temperature? If so, then you can reasonably compare your material to the nearby metals in B/G ratio which are in fact quite brittle for metals. Since this is an important claim of your paper, it warrants more discussion. You also need to throw out reference 15 and replace it with an appropriate citation.

Now, for the more minor issues:

1) There has not been enough review of other cracked based technologies. For example D. Kang, et al., *Nature*, 516, 222-226 (2014) developed a crack based sensor. They conduct a more thorough review of crack based technologies and manipulation, including citing papers which have explicitly investigated the patterning of cracks. This may be a good starting point to provide more information to readers and place your work in context.

2) The device you have developed is clearly a "micro-electromechanical machine" but nowhere is this stated explicitly nor is there any review of other MEMS based memory elements to put the performance of the device into context versus competing technologies. I strongly recommend the authors conduct at least some background in this area for both memory and logic. MEMS often require high electric fields to operate to "pull-in" the counter electrode. Can you comment on how your field of switching compares and how this may translate into a voltage for use in applications? Can you comment on the manufacturing advantages of your system as opposed to conventional MEMS?

3) Along the same lines, a major use of MEMS switches is not for memory, but for radio frequency switches. The figure of merit for RF switches is often defined as the cut-off "frequency" by: $(2\pi C_{off} R_{on})^{-1}$. Based on the described geometry of the gap (300 nm width, 35 nm thick, 100 um long, 10 ohm resistance), I calculate a frequency of: 150 THz, which is very good, though your device is high aspect ratio and I did not account for fringe fields which would reduce this value. The major issue is the gate capacitance, which, in your present geometry, reduces four figure of merit to 60-30MHz. The high dielectric constant of the substrate is also a problem here. Your system also has the added

advantage of nonvolatility which is rare in the area. I recommend you consider analyzing this further and maybe including in the present work, but this is not required for publication.

4) For memory applications, important parameters are the retention, switching speed, switching energy, and endurance. Though it is obviously very good, I do not think the retention performance is explicitly stated. It may be worth citing a work that compares the retention of MEMS to other NVM if one can be found. The switching speed I believe is adequately addressed. For the switching energy, can you provide any estimate of the limit of the switching energy? An estimate of the energies of crack formation in MnPt may suffice. This could come from first principles, or back-of-envelope estimates. For many memory applications, the endurance is very good but you may wish to consider citing a reference stating requirements for certain applications.

5) The results in Figure 4A are quite intriguing, since there is a sharp exponential dependence on the lifetime with frequency. The parasitic of the circuits don't appear large enough to explain this effect so it must be related to the specimen being characterized. My brief look at the literature has yielded a scarcity of frequency dependent information on the crack growth. The theoretical works I found don't discuss frequency dependence either. Is there an explanation for this effect? If it is still under investigation in the literature it would be better for the readers for this to be stated explicitly as an open area of investigation. It's not central to the paper so I don't consider an absence of an explanation to be a barrier to publication.

6) You claim that the Mn deficiency in your films is caused by the low sputtering yield of Mn. This is a claim repeated from your earlier publication, reference 5. However, it's well known that a difference in sputtering yield causes enrichment at the surface of the sputtering target of the lower yield species. This enrichment, leads to increased emission of the lower sputter yield atom until the net flux of atoms from the target is equal to the bulk target composition (see Milton Ohring, the Materials Science of Thin films, Ch. 4). Is the Mn deficiency perhaps caused by some other effect? My understanding was that lighter elements are more strongly scattered to higher angles by the background sputtering gas particles (in this case at 3 mTorr), leading to enrichment of heavier elements which are more weakly scattering. Can you provide an appropriate reference justifying the sputter yield dependence on the composition?

7) Reference 18 does not contain any actual information about the Voigt-Reuss-Hill approach. It's also not clear that the Voigt-Reuss-Hill approach was used Reference 18 since only the Voigt approximation is mentioned.

Response to the Referees' Comments

Reviewer #1

1) This paper reports about a novel device type for current switching on the micro scale, based on opening and closing of micro cracks in a (well-chosen) metal layer activated by a piezoelectric thin film (PMN-PT). The switch works with bipolar gate voltages of interestingly low level (about 1/- 1 V), and exhibits over 4 orders of magnitude on/off current ratio, as required by microelectronic logic applications. The authors also showed a lifetime of 10(+7) cycles. This is quite a good start. The device has the potential to replace a field effect transistor (FET's), however, needing a bipolar voltage input (positive voltage to be in open state, and a negative voltage to be in closed state, or the other way round). FETs have a defined state at zero gate voltage, and just need one gate polarity to change the status.

Response: We greatly acknowledge the referee for her/his time to review our manuscript and the above encouraging comments on the work.

2) The current cannot behave the same way as the ferroelectric polarization. A crack opens as soon as the electric field applied to the PMN-PT exceeds a certain threshold value, which is in the general case not coinciding with the ferroelectric switching (as shown in fig. 2c), but it can. As soon as $E_g > E_c$, the stress (strain) flips to a positive value, which potentially can lead to a cracking in the adjacent conductor film. The loop show in fig. 2c does not correspond to a physically logic picture. The stress at branch 2 is as much tensile as the branch 4, and both should be in the cracked, i.e. low current state. This loop is possibly obtained for a situation where a first tensile stress cycle (no 4 in fig. 2c) was not sufficient to crack the line, but the second (no 1 \diamond 2) was. This was accidentally like this. The next cycle would look differently.

Response: We have to sincerely apologize to the Referee for a lack of clarity in our discussion as to the actual mechanism of crack formation in response to the reversal of the polarization, which would likely be misleading to other readers as well. The referee is absolutely correct that if average strain was to be the only cause, the conductivity on either side of the bias curve should be the same, because in both cases, the average strain state of the crystal is the same. In fact, in both directions, upon application of an electric field, the crystal undergoes a lateral contraction, and the metal film should be in a compressive state, for which cracks would not be expected. However, as we describe in the revised manuscript:

“It is important to remind ourselves that ferroelectric crystals do not switch uniformly but rather through the motion of domain walls, which are partially pinned at surface defects. The local strains occurring near a surface are therefore highly non-uniform, and not symmetric with respect to the reversal of the applied field. In our case, this leads to a switching behavior such that a positive bias (segments 1,2) results in crack opening, whereas a negative bias (segments 3,4) leads to the closing of the cracks, despite the fact that the average strain state of the crystal in both directions is such that one would expect a compressive state – locally, however, domain motion clearly leads to a tensile behavior without which the cracks would not open. Because the metal layer is deposited onto a surface with pre-existing domains and pinning defects, these two

states are highly reproducible and not symmetric with field, contrary to the macroscopic strain in the crystal.”

We have now also clarified the figure caption for Fig 2c, inserting the statement “(data connecting segments 2 and 3 is not shown because it is below the measurement limit)” because the referee seemed to interpret the figure as not having data at 0 V, and believes that subsequent cycles would have looked differently. As the figure caption clearly states, Fig. 2c is actually the superposition of 9 repeated cycles. The switching is so highly reproducible that the reader may mistake it for a single curve, just as the referee might have done. Therefore, we also added the words “Repeated and reproducible switching” to the figure caption.

3) As to memory applications, the suitability of the crack device is more delicate. It is not shown that the cracking is bistable. Only the ferroelectric loop is it (see above). It is not shown what is going on at zero Volt with a crack. The memory effect requires that the state remains unchanged when the gate field is removed. It is clear that when the state is a closed=uncracked state nothing is changing. However, if we deal with an open=cracked state, how can we be sure that there is no migration to close the crack (there can be a negative strain/stress imposed by ferroelectric domains, even without applied field), so that at small reading fields, a closed state appears. Even more difficult is the situation when a cracked line should be reprogrammed for an un-cracked one. How is it possible to heal the crack, and how long does it take? This is not discussed or shown in the manuscript.

Response: We are really sorry for neglecting to discuss this rather important aspect in the manuscript. Based on our experimental experience the non-volatility of the opening cracking state seems “guaranteed” from the very beginning and is also the main starting point. Indeed, we first realized such a switch in the end of 2014 via electrical measurements. To confirm that, we specifically opened cracks in MnPt/PMN-PT heterostructures and shipped the samples to other institutes for repeating measurements and atomic force microscopy imaging. The cracks stayed open until a negative bias was applied.

In addition, we passed several samples with cracks open to Prof. Long You’s device fabrication group in *Huazhong University of Science & Technology* in China for smaller device patterning in May, 2015. Until recently after the submission of this manuscript, we heard back from them and they finally got the device fabrication process correct (this includes the time for setting-up his own lab) and successfully patterned sub-micro cracks.

Fig. R1: Microscopy images of a pattern crack device. **a**, 3D microscopy image of a patterned device based on a 20 nm x 500 nm crack. **b**, Zoom-in SEM image of the crack region of the device.

As shown in Fig. R1, the crack is still open. Also from the two-probe resistance measurements, the initial resistance state across the crack is highly insulating. This means that after 22 months, the cracking state is still there.

Such patterned sub-micro devices are under electrical and endurance testing right now but preliminary results show that they can be well switched back and forth and the data are promising. We would like to further pursue this for a follow-up study.

Fig. R2: Retention of the high-resistance and low-resistance states (HRS & LRS). Each state was monitored for 20 days.

To collect some representative data for describing the non-volatile feature, we have tracked a single crack (~300 nm x 100 um) at the two different resistive states for two months, and the data are shown in Fig. R2. The excellent non-volatility indeed also confirms that the formation of cracks is due to the local strain and deformation at grain boundaries induced by non-uniform domain switching. In contrast, the macroscopic strain in PMN-PT has a symmetric butterfly loop and is volatile at zero-field, for example, see Fig. 1 in Thiele, C. *et al.* Influence of strain on the magnetization and magnetoelectric effect in $\text{La}_{0.7}\text{A}_{0.3}\text{MnO}_3/\text{PMN-PT}$ (001) ($A=\text{Sr}, \text{Ca}$). *Phys. Rev. B* **75**, 054408 (2007).

We have included this new data into the revised manuscript as Fig. 4c with the relevant text:

“Based on our experience on small device fabrication of such heterostructures, the “crack” state is stable for at least 22 months. Representative data for the retention of the high-resistance and low-resistance states (HRS & LRS) collected in two months are shown in Fig. 4c. The excellent non-volatility is consistent with crack formation as a consequence of the local strain and deformation at grain boundaries induced by non-uniform domain switching. In contrast, the macroscopic strain in PMN-PT has a symmetric butterfly loop and is volatile at zero-field [Thiele, C. *et al.* Influence of strain on the magnetization and magnetoelectric effect in $\text{La}_{0.7}\text{A}_{0.3}\text{MnO}_3/\text{PMN-PT}$ (001) ($A=\text{Sr}, \text{Ca}$). *Phys. Rev. B* **75**, 054408 (2007)].”

The referee wonders if small reading fields would change the state of the memory structure. Again, we apologize for failing to be more explicit in the manuscript, which seems have confused the Referee here – in this FET-like device, the switching voltage/field is to be applied to the insulating piezoelectric crystal perpendicularly for “writing”, whereas read-out occurs exclusively through a resistance measurement of the intermetallic film by a rather small in-plane voltage of 0.1 V applied to the intermetallic film, which cannot influence the perpendicular ferroelectric state of the crystal. We therefore add the following sentence: “Finally we note that the writing and reading signals are applied independently of each other, and reading of the device cannot trigger a back-switching behavior.”

Finally, the referee asks whether it is possible to “heal the crack” – clearly, there is an intentional way to close the crack (forward poling); and the crack stays open upon opposite switching for very long time.

4) Fig.4: did the authors make sure that the applied voltage is really applied on the device? An RC problem might reduce the amplitude at higher frequency.

Response: Indeed, at all frequencies we listed in Fig. 4a, we observed the “opening” and “closing” of cracks. This increases our confidence on this issue, *i.e.*, it was not an RC problem.

5) Finally one can say that the shown element can be used as a switch, useful for logic applications. However, the memory case is doubtful. Would need more convincing data on the evolution of the cracked state at zero field and reading field, and a concept for closing the crack. The authors should reduce their claims about the memory, and consider correctly the relation between stress and polarization (see figure in pdf file).

Response: We hope that with the answers provided to comment 3) of the referee, the concern expressed here is also addressed. We observe repeatable switching (true switching upon pulses with opposite polarity, as shown in Fig. 4b) of more than 10 million cycles for which the cracks opened and closed, with an 8 order of magnitude modulation of the resistance. Our clarified statement that the data at 0 V (*i.e.*, the data connecting segments 2 and 3) is below the measurement limit and the retention data should answer the referee's request for "more convincing data on the evolution of the cracked state at zero field and reading field".

The referee has provided us both constructive and critical comments, which have been all helpful for improving the readability of our manuscript and the soundness of the research. We therefore would like to sincerely thank the referee again for her/his thinking our on work in the end of this part.

Reviewer #2

This is very nice piece of well written work with a clear objective and convincing results. I have no scientific questions. The only missing part is the electromechanical modelling of the reversible crack opening and closure.

Response: We sincerely thank the referee for reading our manuscript and also the above positive comments on our writing and experimental results.

Motivated by the referee's illuminating suggestion, we have collaborated with Prof. Long-Qing Chen's group at the Pennsylvania State University and performed the phase-field modeling on the reversible crack opening and closing.

After six months, we have successfully simulated the crack opening and closing. Basically, we propose a possible mechanism for the strain-mediated electric-field controlled reversible crack opening and closure. A simplified two-domain configuration is considered, in which one domain can be switched by an electric field and the other is pinned. As shown in Figure 3c and 3d, electric-field-induced 109° polarization switching in the switchable domain can generate a large pulsed strain of 0.2% (calculated based on the lattice parameters of the rhombohedral PMN-PT). When the strain is ON, the crack forms; when the strain is OFF, the crack closes. As the electric-field 109° polarization switching has been experimentally shown to be both reversible and non-volatile (see Zhang, et al. Phys. Rev. Lett. 108, 137203(2012)), the associated crack opening and closure should also be reversible and non-volatile. We validated this hypothesis through phase-field modeling of crack evolution (see Figure 3e). The results show that the crack can stably exist along the domain wall when the strain is ON, and will gradually vanish when the strain is OFF. Detailed discussions and methods have been added to the revised manuscript.

As the objective of the manuscript is more an engineering issue this might be not necessary. My main concern is whether the manuscript is suitable for Nature Communications. As I am not working in the semiconductor storage area it is difficult for me to judge that and to estimate the impact of this investigation for this community. By trend I would see it more appropriate in a more engineering related journal.

Response: Such devices utilizing mechanical and electrical properties of intermetallic films and cracks in ferroelectrics would not only be interesting to the communities of intermetallics, and ferroelectrics, but more importantly, the resulting heterostructures can work as field-effect transistors, non-volatile memory, and also microelectromechanical systems. As the referee might be able to see the comments from Referee #1 and Referee #3 in this response letter, who we firmly believe are well-established experts in the semiconductor storage area, both of them had positive thoughts on the *impact* of this work.

The referee is definitely right that lots of engineering issues would be involved for pushing such devices for practical applications. However, we believe that the first propose and demonstration of such a novel device based on cracks deserve more attention from researchers in a broad range of communities.

Reviewer #3

In their paper, Electrically Reversible Cracks in Intermetallic Film controlled by an Electric Field the authors study the reversible changes in a MnPt film deposited on a PMN-PT substrate. Such a substrate is known to undergo elastic and inelastic deformation in the presence of an electric field. The authors discover that the former effect, elastic deformation, allows for the reversible transfer of the latter, inelastic deformations (in this case cracks), from the substrate material to the film.

This discovery deserves a great deal of credit and I greatly enjoyed reading the paper and it is my pleasure to recommend it for publication. This is truly a novel switch and the provided results are both convincing and sufficiently well described to allow reproduction by other researchers. However, though the fundamental observations of the paper are sound and I think at a minimum this will provoke people to think much more broadly about the types of possible memories and switches can be achieved.

Response: We greatly appreciate the referee's highly encouraging comments on our work. Frankly speaking, it is kind of unexpected but rather impressive for all of us that we have learned a lot of knowledge, some of which was out of our knowledge base or even textbooks, via this referee's extremely detailed comments and suggestion. We are so grateful to such a knowledgeable referee that she/he has spent a great amount of time for reading our manuscript, investigating literatures spanning over a long period regarding this work, and providing her/his own enlightening thoughts to us, which are rather helpful for improving this work.

There is at least one serious issue that must be addressed before the paper can be considered acceptable for publication. There are likewise a few minor issues that must be addressed as well as some further investigation which I encourage the authors to do to strengthen in the claims in the paper.

Firstly, the serious issue concerns the so called "critical value" of $B/G = 1.75$. Quite frankly, until I read this paper, I've never heard of it. It's not mentioned in my texts on fracture

mechanics. The reference you have provided, reference 15, is a completely inappropriate reference since it is neither the primary source of this “critical value” or an authoritative reference (such as a review or textbook). Reference 15, however, does apparently cite the origin of this “critical value” and it appears to be from a seminal work by Pugh from 1954.

Modern references constantly talk about “Pugh’s criterion” for a ductile-brittle transition based on the B/G ratio and a “critical value”, but nowhere in Pugh’s 1954 paper does he ever formulate this criterion. The paper isn’t about a universal ductile-brittle number, instead, it’s about comparing the ductility of similar materials by comparing their relative B/G value. The critical value you reference (sometimes stated as 1.74, 1.75, and otherwise 2) is the B/G ratio of iridium, which happens to be the least ductile in the family of “elemental f.c.c. metals of high melting point” as he calls them in his paper.

This is important, because, as is often discussed in texts in fracture mechanics, many metals, particularly BCC metals, will undergo a brittle-ductile transition with temperature. This is because ductility is determined by the number of active slip systems which allow for the motion of dislocations (in many metals, this depends on temperature). As Pugh makes a point of noting, the B/G ratio doesn’t work if you try to compare materials with vastly different melting temperatures. Zinc, for example, has a much lower melting temperature than the “elemental f.c.c. metals of high melting point” and so therefore has a much higher homologous temperature. It’s B/G ratio is 1.59, below the so-called universal “critical value” and, according to Pugh, undergoes 25% elongation before fracture during tensile testing. That makes zinc decidedly ductile and Pugh explicitly limits the validity of his model to elemental metals at 1/3 their homologous temperature. Some authors appear to include this caution explicitly when citing Pugh.

The scope of Pugh’s model is further limited by the fact that it doesn’t even work for all “elemental f.c.c. metals of high melting point” outside of the transition metals. Thorium meets all the limitations of the model with a B/G ratio of 1.74 (in Pugh’s paper), but it undergoes up to almost 50% elongation before fracture. This is similarly true for uranium (though it is orthorhombic). It’s B/G ratio is about 1 and is also quite ductile at room temperature.

I have tried to determine the origin of this “critical value” formalism as you use it. In the literature, and the earliest reference I can find is: J. R. Morris, et al., *Acta Materialia*, 52, 4849-4857 (2004). This paper was published in 2004, nearly 50 years after Pugh’s criterion was formulated! This paper discusses Pugh’s work in better detail and do take steps to limit the scope of critical value to “FCC metals with melting temperatures greater than 900C,” but later authors (mostly users of DFT) appear to be continuously expanding the scope of Pugh’s criterion to cover almost all materials without critical evaluation as to its origin - this has caused me a great deal of confusion.

If you can disprove this by providing an authoritative reference (preferably one published before 2004) or by showing where in Pugh’s paper he proves that this is a generalizable critical value for all materials, I would be happy to defer. However, if you cannot do this, I recommend you reformulate this section based on Pugh’s original criteria – that is, among similar materials below 1/3 their homologous temperature, the relative B/G ratio can act as a predictor of brittleness. In which case, since the L10 structure is an FCC derivative and Mn/Pt are both transition metals, you could reasonably compare MnPt to the other “high melting point FCC metals” if it also has a

high melting point. Do your theoretical calculations also provide an estimate of the melting temperature? If so, then you can reasonably compare your material to the nearby metals in B/G ratio which are in fact quite brittle for metals. Since this is an important claim of your paper, it warrants more discussion. You also need to throw out reference 15 and replace it with an appropriate citation.

Response: We fully agree with the referee on this aspect and would like to thank the referee for pointing this out.

The B/G ratio is a simple model to judge the brittleness or ductility of a solid material, since the bulk modulus B can be considered as the resistance to fracture and the shear modulus G the resistance to plastic deformation. The critical B/G ratio (1.75 or 1.74) to separate the brittle and ductile materials was first implied by Pugh (in 1954)¹ for fcc metals with high melting point (page 838, section 4.3 in his paper), Pugh indicated that the B/G values of these fcc metals vary from 1.74 for Ir (a brittle phase with a very small elongation) to 6.14 for Au (50% elongation). In fact, the microscopic origin of this empirical parameter can be understood from an isotropic cubic crystal with Cauchy relations $C_{11}=3C_{12}$ and $C_{12} = C_{44}$, resulting in a B/G value of 1.67².

We agree with the Reviewer that the critical value of $B/G = 1.75$ is mainly used in the first-principles community for various materials which may be incorrect for some cases. Based on the best of our knowledge, we knew Ravindran *et al.* (in 1998)² used this value to judge the ductility of orthorhombic TiS_2 by considering the ductile Ti ($B/G = 2.47$) and the brittle Si ($B/G = 1.49$). However, we also do not know the first researcher(s) who used the $B/G = 1.75$ as Pugh's criterion.

In addition to the exceptions (beyond the critical $B/G = 1.75$) pointed out by the Reviewer (Zn and Th), we³ also found that the brittle NbCr_2 has a large B/G ratio of 3.69. A review article by Gandhi and Ashby (in 1979)⁴ indicated that the trend of Pugh's criterion is right, but the criterion does not provide a satisfactory indicator for brittleness by examining the *fcc*, *bcc*, and *hcp* metals, and ceramics.

Following the Reviewer's suggestion, we examined the B/G ratios for 24 compounds with the $L1_0$ structure (space group $P4/mmm$) in terms of the Materials Project database⁵, see the new Table S4 in supplementary information. The B/G ratio of these $L1_0$ -type compounds varies from 1.58 to 4.82 with an average around 2.8. In particular, the brittle TiAl ⁶ has a $B/G = 1.64$, and the ductile CoPt ⁷ has a $B/G = 2.00$. Similar to the *fcc* metals with high melting point, $B/G = 1.75$ should be a reasonable value to identify the brittle and ductile materials with $L1_0$ -type structure. We further noticed that *fcc* Mn (a metastable phase) and *fcc* Pt have B/G ratios of 2.14 and 4.98, respectively (see the new Table S4 in supplementary information). The present MnPt has a $B/G = 1.84$, indicating MnPt is near-brittle (or near the borderline of brittleness) within the $L1_0$ -type compounds. And the brittleness of MnPt stems from the antiferromagnetic Mn atoms by considering the B/G ratios for the FM and AF MnPt (2.32 vs. 1.84, see the new Table S4 in supplementary information).

Finally, we would like to comment that the theoretical calculations can predict the melting temperature based on such as molecular dynamics simulations⁸, but these simulations are beyond the scope of the present work.

Response: In the revised manuscript, we have removed the previous ref. 15, and used Pugh's paper¹ as a new reference. In the main text of the revised manuscript, we wrote that:

“This yields a bulk/shear modular ratio $B/G \approx 1.84$, indicating MnPt is likely to be brittle due to the antiferromagnetic Mn atoms according to Pugh's criterion¹⁵, see details in supplementary information.”

And in the supplementary information, we added one sub-section “**Pugh's criterion (B/G ratio) to judge brittleness/ductility**” together with a new Table S4:

"The Pugh's criterion based on the B/G ratio is a simple model to judge the brittleness or ductility of a solid material, since the bulk modulus B can be considered as the resistance to fracture and the shear modulus G the resistance to plastic deformation. A critical B/G ratio (around 1.75) to separate the brittle and ductile materials was implied by Pugh¹ based on *fcc* metals with high melting point. The microscopic origin of this empirical parameter ($B/G = 1.75$) can be understood from an isotropic cubic crystal with Cauchy relations $C_{11} = 3C_{12}$ and $C_{12} = C_{44}$, resulting in a B/G value of 1.67 (Ref. 2). The critical value ($B/G = 1.75$) has been widely used in first-principles community for various materials, see such as the work done by Ravindran *et al.*² in 1998 for the orthorhombic TiS_2 by considering the ductile Ti ($B/G = 2.47$) and the brittle Si ($B/G = 1.49$).

Although the criterial B/G ratio (1.75) was driven from *fcc* metals with high melting point, we found that it works for $L1_0$ -type compounds by considering the brittle TiAl ⁶ with $B/G = 1.64$ and the ductile CoPt ⁷ with $B/G = 2.00$ (data from the Materials Project database⁵, see Table S4). It is worth mentioning that the criterial $B/G = 1.75$ don't work for all materials as pointed by, for example, Pugh¹, Gandhi and Ashby⁴, and Long *et al.*¹.

The present MnPt with $B/G = 1.84$ should be near-brittle within the category of $L1_0$ -type compounds, and MnPt is more brittle than its constituent elements, *i.e.*, the metastable *fcc* Mn with $B/G = 2.14$ and *fcc* Pt with $B/G = 4.98$, see Table S4. It is further noticed that the brittleness of MnPt stems from the antiferromagnetic Mn atoms by considering the B/G ratios for the FM and AFM MnPt (2.32 vs. 1.84, see Table S4)."

Now, for the more minor issues:

1) There has not been enough review of other cracked based technologies. For example D. Kang, et al., Nature, 516, 222-226 (2014) developed a crack based sensor. They conduct a more thorough review of crack based technologies and manipulation, including citing papers which have explicitly investigated the patterning of cracks. This may be a good starting point to provide more information to readers and place your work in context.

Response: This is very useful. We have added relevant discussion on previous crack-based technologies in the beginning of the 3rd paragraph.

2) The device you have developed is clearly a “micro-electromechanical machine” but nowhere is this stated explicitly nor is there any review of other MEMS based memory elements to put the performance of the device into context versus competing technologies. I strongly recommend the authors conduct at least some background in this area for both memory and logic. MEMS often require high electric fields to operate to “pull-in” the counter electrode. Can you comment on how your field of switching compares and how this may translate into a voltage for use in applications? Can you comment on the manufacturing advantages of your system as opposed to conventional MEMS?

Response: Yes, it is indeed a MEMS realized through ferroelectric oxides. The typical pull-in voltage for conventional MEMS is 3~8 V [Gaddi, R. *et al.* MEMS technology integrated in the CMOS back end. *Microelectron. Reliab.* **50**, 1593-1598 (2010)]. The electric field applied onto the insulating PMN-PT for switching in our devices is ~0.83 kV/cm, which corresponds to ~4.1 V even if we thin PMN-PT crystals down to 50 μm , which would be comparable to the operation voltage of other MEMS. Moreover, such crack formation is expected to work even for 1- μm -thick PMN-PT films [Baek, S. H. *et al.* Giant piezoelectricity on Si for hyperactive MEMS. *Science* **334**, 958-961 (2011)], which would result in a much lower operation voltage ~80 mV.

Conventional MEMS consists of multiple layers and require rather complicated and thus expensive fabrication processes, which are typically volatile at zero voltage. The single layer film structure of our devices is much simpler and the fabrication is much easier. In addition, they are non-volatile. We have emphasized these advantages over conventional MEMS into the revised manuscript.

3) Along the same lines, a major use of MEMS switches is not for memory, but for radio frequency switches. The figure of merit for RF switches is often defined as the cut-off “frequency” by: $(2\pi C_{\text{off}} R_{\text{on}})^{-1}$. Based on the described geometry of the gap (300 nm width, 35 nm thick, 100 μm long, 10 ohm resistance), I calculate a frequency of: 150 THz, which is very good, though your device is high aspect ratio and I did not account for fringe fields which would reduce this value. The major issue is the gate capacitance, which, in your present geometry, reduces four figure of merit to 60-30MHz. The high dielectric constant of the substrate is also a problem here. Your system also has the added advantage of nonvolatility which is rare in the area. I recommend you consider analyzing this further and maybe including in the present work, but this is not required for publication.

Response: We greatly thank the referee for bringing this new insight to us as we did not even think about that before. Instead of including this rather attractive aspect for our preliminary model devices which are not well patterned, we would like to intensively pursue the figure of merit in our follow-up study based on well-patterned much smaller devices.

4) For memory applications, important parameters are the retention, switching speed, switching energy, and endurance. Though it is obviously very good, I do not think the retention performance is explicitly stated. It may be worth citing a work that compares the retention of MEMS to other NVM if one can be found. The switching speed I believe is adequately addressed. For the switching energy, can you provide any estimate of the limit of the switching energy? An

estimate of the energies of crack formation in MnPt may suffice. This could come from first principles, or back-of-envelope estimates. For many memory applications, the endurance is very good but you may wish to consider citing a reference stating requirements for certain applications.

Response: **Referee 1** also mentioned the retention issue. It is our fault that we missed to discuss the retention aspect in the manuscript. Based on our experience on small device fabrication of such heterostructures, the “crack” state is stable for at least 22 months ($\sim 10^7$ s). We have collected some representative data during these days and added them into the revised manuscript.

The switching speed and the switching energy are mainly determined by the polarization switching in ferroelectric oxides. For example, as we referred to [J. Li, B. Nagaraj, H. Liang, W. Cao, Chi. H. Lee, and R. Ramesh, *Appl. Phys. Lett.* 84, 1174-1176 (2004)] in the manuscript, the polarization switching could be as fast as ~ 100 ps. The switching energy can be estimated by $E_{\text{Switching}} = 1/2 P_s V S$, where P_s is the saturation ferroelectric polarization ($\sim 25 \mu\text{C}/\text{cm}^2$ for PMN-PT), V is the switching voltage and S is the cell area. If such a crack device can be scaled down to a footprint area of $100 \times 100 \text{ nm}^2$, with PMN-PT thickness of $1 \mu\text{m}$ (in terms of films), the switching energy for each memory bit/cell would be $E_{\text{Switching}} = 1/2 P_s V S = 12.5 \mu\text{C}/\text{cm}^2 \times 0.83 \text{ kV}/\text{cm} \times 1 \mu\text{m} \times 100 \text{ nm} \times 100 \text{ nm} \approx 0.1 \text{ fJ}$, which is one order of magnitude smaller than the energy consumption of non-volatile phase change memory ($\sim 1 \text{ fJ}/\text{bit}$, ITRS 2013) and three orders of magnitude lower than that of the spin-torque-transfer RAM ($0.1 \text{ pJ}/\text{bit}$, ITRS 2013). We have added such estimation into the revised manuscript. It is therefore promising for low-energy memory applications.

We have spent six months in performing the phase-field simulation on the crack formation and it turns out that the electric-field induced crack formation and closure is mainly governed by the competing electromechanical-strain-related elastic energy and the surface energy. The higher the surface energy, the harder the crack formation (*i.e.*, the higher the crack formation energy). In our simulation, we found that while the elastic energy is higher than $10 \text{ MJ}/\text{m}^3$, the crack will form.

In addition, we have cited [Yang, J. J., Strukov, D. B. & Stewart, D. R. Memristive devices for computing. *Nature Nano.* 8, 13-24 (2013)] to justify that the endurance of our devices are good for memory applications as the endurance of commercial flash memory is $\sim 10^4$ cycles.

5) The results in Figure 4A are quite intriguing, since there is a sharp exponential dependence on the lifetime with frequency. The parasitic of the circuits don't appear large enough to explain this effect so it must be related to the specimen being characterized. My brief look at the literature has yielded a scarcity of frequency dependent information on the crack growth. The theoretical works I found don't discuss frequency dependence either. Is there an explanation for this effect? If it is still under investigation in the literature it would be better for the readers for this to be stated explicitly as an open area of investigation. It's not central to the paper so I don't consider an absence of an explanation to be a barrier to publication.

Response: To be honest, we do not really have a good physical understanding on the frequency-dependent lifetime either. As suggested by the referee, we have acknowledged this open question in the revised manuscript.

6) You claim that the Mn deficiency in your films is caused by the low sputtering yield of Mn. This is a claim repeated from your earlier publication, reference 5. However, it's well known that a difference in sputtering yield causes enrichment at the surface of the sputtering target of the lower yield species. This enrichment, leads to increased emission of the lower sputter yield atom until the net flux of atoms from the target is equal to the bulk target composition (see Milton Ohring, the Materials Science of Thin films, Ch. 4). Is the Mn deficiency perhaps caused by some other effect? My understanding was that lighter elements are more strongly scattered to higher angles by the background sputtering gas particles (in this case at 3 mTorr), leading to enrichment of heavier elements which are more weakly scattering. Can you provide an appropriate reference justifying the sputter yield dependence on the composition?

Response: This is a very good point. Our statement on the low sputtering yield may not be correct.

In our experiments, the sputtering of MnPt always generates a lower Mn ratio in thin films regardless of the sputtering power and Ar pressure we used. We tried lots of conditions for optimizing the thin film composition towards Mn₅₀Pt₅₀ based on a Mn₅₀Pt₅₀ target, but all the films had a Mn deficiency of 4-6%. Such an experimental fact seems to exist in early studies such as [Severin, C. S. & Chen, C. W. Ferromagnetic behavior of disordered MnPt films produced by rf sputtering. *J. Appl. Phys.* **49**, 1693-1695 (1978)], where the authors mentioned "Back sputtering was also used to compensate for the lower sputtering rate of Mn."

For single Mn and Pt targets, the sputtering yield of Mn is lower. For example, with 600 eV argon ions in a typical argon plasma, the sputtering yield is 1.3 and 1.6 (atoms/ion) for Mn and Pt, respectively (<http://www.semicore.com/reference/sputtering-yields-reference>). A lower sputtering yield of an element in a binary alloy target has two opposite effects: deficiency in sputtered films and richness on the target surface. As a result, there is a composition self-correction effect in sputtered thin films. After we read about the sputtering effects on a binary alloy target surface in the classical book "the Materials Science of Thin Films" authored by Milton Ohring, we agree that the sputtering yield argument may not be responsible for the off-stoichiometry in sputtered films as a *full* composition correction will be achieved after a few hundred atom layers are sputtered from a binary alloy target surface.

But what was not considered in Milton's book is the angular distribution of sputtered atoms as pointed out by the Referee, which depends on the energy of incident Ar ions, the mass of atoms and the target's crystallography. As Mn and Pt have a large difference in mass, their angular distribution should be different, which may have contributed to the Mn deficiency in sputtered films. We have revised the statement as follows:

"The Mn deficiency may be because the lighter Mn atoms are more strongly scattered to higher angles by the background Ar atoms."

The sputtering yield dependence on composition was investigated for both nonmixable binary alloy systems and solid solutions in [Betz, G. Alloy sputtering. *Surf. Sci.* **92**, 283-309 (1980)], which could be a useful reference.

7) Reference 18 does not contain any actual information about the Voigt-Reuss-Hill approach. It's also not clear that the Voigt-Reuss-Hill approach was used Reference 18 since only the Voigt approximation is mentioned.

Response: Thanks for pointing out this less accurate reference.

The Hill approach was used to study the bulk modulus, shear modulus, and other elastic properties. A new reference has been provided for the Hill approach [Hill, R. The elastic behaviour of a crystalline aggregate. *Proc. Phys. Soc. A* 65 (1952) 349-354], which gives the average of the Voigt and Reuss results.

In the revised manuscript, the last sentence in the Method section was revised as follows:

"The single-crystal elastic constants of MnPt were predicted by the strain-stress method based on first-principles calculations⁹, and the elastic properties for polycrystalline MnPt were estimated using the Hill approach¹⁰."

References:

1. Pugh, S. F. XCII. Relations between the elastic moduli and the plastic properties of polycrystalline pure metals. *Philos. Mag.* **45**, 823–843 (1954).
2. Ravindran, P. *et al.* Density functional theory for calculation of elastic properties of orthorhombic crystals: Application to TiSi₂. *J. Appl. Phys.* **84**, 4891–4904 (1998).
3. Long, Q. *et al.* C₁₅ NbCr₂ Laves phase with mechanical properties beyond Pugh's criterion. *Comput. Mater. Sci.* **121**, 167–173 (2016).
4. Gandhi, C. & Ashby, M. F. Fracture mechanism maps for materials which cleave: F.C.C., B.C.C. and H.C.P. metals and ceramics. *Acta Metall.* **27**, 1565–1602 (1979).
5. Jain, A. *et al.* Commentary: The Materials Project: A materials genome approach to accelerating materials innovation. *APL Mater.* **1**, 11002 (2013).
6. Booth, A. S. & Roberts, S. G. The brittle-ductile transition in γ -TiAl single crystals. *Acta Mater.* **45**, 1045–1053 (1997).
7. Greenberg, B. A. *et al.* Optimised mechanical properties of ordered noble metal alloys. *Platin. Met. Rev.* **47**, 46–58 (2003).
8. Jin, Z. H., Gumbsch, P., Lu, K. & Ma, E. Melting mechanisms at the limit of superheating. *Phys. Rev. Lett.* **87**, 55703 (2001).
9. Shang, S. L., Wang, Y. & Liu, Z. K. First-principles elastic constants of α - and θ -Al₂O₃. *Appl. Phys. Lett.* **90**, 101909 (2007).
10. Hill, R. The elastic behaviour of a crystalline aggregate. *Proc. Phys. Soc. Sect. A* **65**, 349–354 (1952).

Summary of changes

Authors:

- We have added Dr. Jiamian Hu, Mr. Yanzhou Ji and Prof. Long-Qing Chen as co-authors due to their contribution to the phase field simulation.
- We have added our students Mr. Han Yan and Mr. Zexin Feng as co-authors due to their contribution to the collection of the retention data.
- We have added Prof. Long You as co-author due to his contribution to the device fabrication work, which helps to clarify the excellent non-volatility of the crack state in this work.

Main Text:

- Changes in the main text are marked in dark blue in the revised manuscript.

Figures:

- In response to Referee #1, we have added the retention data as Fig. 4c.
- In response to Referee #2, we have added the theoretical modeling of the “opening” & “closing” of the cracks in Figure 3.

References:

- In response to Referee #1, we added [Thiele, C. et al. Influence of strain on the magnetization and magnetoelectric effect in $\text{La}_{0.7}\text{A}_{0.3}\text{MnO}_3/\text{PMN-PT}$ (001) (A=Sr, Ca). *Phys. Rev. B* 75, 054408 (2007)] as a new reference for discussing microscopic and macroscopic strain in ferroelectrics.
- In response to Referee #2, we added the references related to the phase field simulation.
- In response to Referee #3, we added [Pugh, S. F. XCII. Relations between the elastic moduli and the plastic properties of polycrystalline pure metals. *Philos. Mag.* 45, 823–843 (1954) & Hill, R. The elastic behaviour of a crystalline aggregate. *Proc. Phys. Soc. A* 65, 349-354 (1952)] as new references and removed the previous Ref. 15.
- In response to Referee #3, we added relevant references regarding crack-based technologies, MEMS and memory applications.

Supplemental material:

- In response to Referee #3, we have added the detailed methods of theoretical calculations and the discussion of B/G ratio, including a sub-section and a new Table S4, and also the mechanical testing results of MnPt films.

Reviewers' comments:

Reviewer #1 (Remarks to the Author):

The authors answered well to all the comments, and questions of the reviewers, and improved the manuscript adequately. I accept their explanation of asymmetric crack-switching due to pinned domains. Indeed, in a rhombohedral PMN-PT with (001) orientation, strong shear stresses are expected to be created for each domain, because electric field and polarization are not parallel. When these shear stresses have opposite sign on neighboring domains, in-plane forces act on vertical (110)-type domain walls (separating 109° domains), able to open the (110) domain wall as a crack. The next point is then how to organize domain pinning for a device, but this we can leave to a next paper, presumably. I recommend to accept the article as it is now.

Reviewer #2 (Remarks to the Author):

There were a lot of very detailed and serious questions raised by the reviewers and I am impressed by the honest response and detailed answers provided by the authors. Concerning my questions I am happy to see that the authors included phase field modelling, which helps to understand the crack developing mechanism as the explanation given in the original manuscript was misleading. Having read the manuscript again I strongly recommend nature communications to publish this manuscript.

Reviewer #3 (Remarks to the Author):

It is my pleasure to recommend this paper again for publication since the authors have done very good work addressing almost all of my requests.

In particular, the analysis of Pugh's Criterion, which has been lacking for some time, will hopefully add clarity to researchers in the field.

The authors have adequately addressed the other minor issues I have raised from including necessary citations, fixing mis-citations, adding important context, and correcting a minor error with regards to sputter yield effects on stoichiometry.

There is however one final error that must be corrected, which is in the analysis on the energy required for switching. In my original comment, I proposed that they consider the energy required to form a crack in MnPt. However, the authors have proposed that the energy required for switching is the energy needed to switch the polarization of the PMN-PT. This is not strictly true, since the elastic energy from switching the polarization and bending the PMN-PT will only dominate for large sizes. As the device is scaled, the bonding energy of the interface will dominate the switching.

So yes, the elastic energy of switching the polarization is 0.1 fJ. However, the energy required to form a new interface in the MnPt is simply the surface energy times the area of the crack.

In our case, surface energies are usually around 1 J/m^2 , the film thickness is 35 nm, and the crack length is 100 nm. Since there are two interfaces we then have:

$$2 \times 1 \text{ J/m}^2 \times 35 \times 10^{-9} \text{ m} \times 100 \times 10^{-9} \text{ m} = 7 \times 10^{-15} \text{ J} \text{ or } 7 \text{ fJ}$$

which is quite a bit larger than the proposed 0.1 fJ.

This energy could be larger or smaller depending on the details of the surface energy and the newly formed grain boundary energy after the first breaking and closing cycle.

This is only the energy for forming the crack. Re-forming the crack is possibly a much more energetic process, since breaking the crack likely leads to roughness in the interface. Plastic deformation, which is much higher energy than the energy of crack formation by up to a factor of 1000, is likely necessary to form an intimate interface. E.g., plastic deformation is often used to form high quality seals, as in copper gaskets for UHV systems. The energy barrier to reforming the crack is presently not clear nor can it be easily resolved I think at present. It's also implied by this analysis that there are potentially complex relationships governing energy, ductility, and endurance that the authors will hopefully continue to explore.

Response to the Referees' Comments

Reviewer #1

The authors answered well to all the comments, and questions of the reviewers, and improved the manuscript adequately. I accept their explanation of asymmetric crack-switching due to pinned domains. Indeed, in a rhombohedral PMN-PT with (001) orientation, strong shear stresses are expected to be created for each domain, because electric field and polarization are not parallel. When these shear stresses have opposite sign on neighboring domains, in-plane forces act on vertical (110)-type domain walls (separating 109° domains), able to open the (110) domain wall as a crack. The next point is then how to organize domain pinning for a device, but this we can leave to a next paper, presumably. I recommend to accept the article as it is now.

Response: We thank the referee for recommending publication of our manuscript.

Reviewer #2

There were a lot of very detailed and serious questions raised by the reviewers and I am impressed by the honest response and detailed answers provided by the authors. Concerning my questions I am happy to see that the authors included phase field modelling, which helps to understand the crack developing mechanism as the explanation given in the original manuscript was misleading. Having read the manuscript again I strongly recommend nature communications to publish this manuscript.

Response: We sincerely thank the referee for appreciating our effort for the revision and strongly recommending our work for publication in Nature Communications.

Reviewer #3

It is my pleasure to recommend this paper again for publication since the authors have done very good work addressing almost all of my requests.

In particular, the analysis of Pugh's Criterion, which has been lacking for some time, will hopefully add clarity to researchers in the field.

The authors have adequately addressed the other minor issues I have raised from including necessary citations, fixing mis-citations, adding important context, and correcting a minor error with regards to sputter yield effects on stoichiometry.

Response: We really appreciate the referee for recognizing our effort during the last revision. We would also like to thank the referee for the consistent support to recommend our work for publication.

There is however one final error that must be corrected, which is in the analysis on the energy required for switching. In my original comment, I proposed that they consider the energy required to form a crack in MnPt. However, the authors have proposed that the energy required for switching is the energy needed to switch the polarization of the PMN-PT. This is not strictly true, since the elastic energy from switching the polarization and bending the PMN-PT will only dominate for large sizes. As the device is scaled, the bonding energy of the interface will dominate the switching.

So yes, the elastic energy of switching the polarization is 0.1 fJ. However, the energy required to form a new interface in the MnPt is simply the surface energy times the area of the crack.

In our case, surface energies are usually around 1 J/m^2 , the film thickness is 35 nm, and the crack length is 100 nm. Since there are two interfaces we then have: $2 \times 1 \text{ j/m}^2 \times 35 \times 10^{-9} \text{ m} \times 100 \times 10^{-9} \text{ m} = 7 \times 10^{-15} \text{ J}$ or 7fJ which is quite a bit larger than the proposed 0.1 fJ. This energy could be larger or smaller depending on the details of the surface energy and the newly formed grain boundary energy after the first breaking and closing cycle.

This is only the energy for forming the crack. Re-forming the crack is possibly a much more energetic process, since breaking the crack likely leads to roughness in the interface. Plastic deformation, which is much higher energy than the energy of crack formation by up to a factor of 1000, is likely necessary to form an intimate interface. E.g., plastic deformation is often used to form high quality seals, as in copper gaskets for UHV systems. The energy barrier to reforming the crack is presently not clear nor can it be easily resolved I think at present. It's also implied by this analysis that there are potentially complex relationships governing energy, ductility, and endurance that the authors will hopefully continue to explore.

Response: We fully agree with the referee on the energy consumption when the device is scaled down. We failed to realize the surface energy part when the crack is newly opened and oversimplified the energy consumption in the previous revised manuscript.

Based on the referee's detailed suggestion, we have now clarified the energy consumption for small devices as the referee calculated.

We also agree that there are complex relationships governing energy, ductility, and endurance, which would be very interesting to pursue in our following studies.

Summary of changes

Main Text:

- In response to **Referee 3**, we re-estimate the energy consumption for small devices based on the referee's suggestion and the changes are marked in dark blue in the last second paragraph of the revised manuscript.
- To comply with the format requirements, we added subheadings as required.

REVIEWERS' COMMENTS:

Reviewer #3 (Remarks to the Author):

The authors have adequately addressed my final concern governing their estimate of the energy of switching. I recommend that the paper be published without further modification.

Brian D. Hoskins, NIST